# Human Sperm Head Vacuoles Are Related to Nuclear-Envelope Invaginations

**DOI:** 10.3390/ijms241210027

**Published:** 2023-06-12

**Authors:** María José Gómez-Torres, Javier Luna-Romero, Pedro José Fernández-Colom, Jon Aizpurua, Manuel Avilés, Alejandro Romero

**Affiliations:** 1Departamento de Biotecnología, Facultad de Ciencias, Universidad de Alicante, 03690 Alicante, Spain; javierlr91@gmail.com (J.L.-R.);; 2Cátedra Human Fertility, Universidad de Alicante, 03690 Alicante, Spain; 3Departamento de Ginecología (Medicina Reproductiva), Hospital Universitario y Politécnico La Fe, 46026 Valencia, Spain; 4IVF Spain, Reproductive Medicine, 03540 Alicante, Spain; 5Departamento de Biología Celular e Histología, Facultad de Medicina, Universidad de Murcia IMIB-Arrixaca, Campus Mare Nostrum, 30100 Murcia, Spain

**Keywords:** human sperm, vacuoles, ultrastructure, morphology, immunocytochemistry, TEM

## Abstract

Nuclear vacuoles are specific structures present on the head of the human sperm of fertile and non-fertile men. Human sperm head vacuoles have been previously studied using motile sperm organelle morphology examination (MSOME) and their origin related to abnormal morphology, abnormal chromatin condensation and DNA fragmentation. However, other studies argued that human sperm vacuoles are physiological structures and consequently, to date, the nature and origin of the nuclear vacuoles remains to be elucidated. Here, we aim to define the incidence, position, morphology and molecular content of the human sperm vacuoles using transmission electron microscopy (TEM) and immunocytochemistry techniques. The results showed that ~50% of the analyzed human sperm cells (*n* = 1908; 17 normozoospermic human donors) contained vacuoles mainly located (80%) in the tip head region. A significant positive correlation was found between the sperm vacuole and nucleus areas. Furthermore, it was confirmed that nuclear vacuoles were invaginations of the nuclear envelope from the perinuclear theca and containing cytoskeletal proteins and cytoplasmic enzyme, discarding a nuclear or acrosomal origin. According to our findings, these human sperm head vacuoles are cellular structures originating from nuclear invaginations and contain perinuclear theca (PT) components, allowing us to define a new term of ‘nuclear invaginations’ rather than ‘nuclear vacuoles’.

## 1. Introduction

Mammalian spermatozoa are highly specialized cells that suffer relevant morphological changes during spermiogenesis. In this late phase of spermatogenesis, several complex events occur that affect the sperm plasma membrane and are involved in chromatin condensation, acrosome and tail genesis, cytoplasm extrusion and biochemical remodeling [1]. In this context, it is important to note the development of specialized cytoskeletal head components which do not appear in any other cell type [2,3]. The subacrosomal region (SAR) cytoskeleton is located between the inner acrosomal membrane and the nuclear envelope. The postacrosomal region (PAR) cytoskeleton is located between the plasma membrane and the nuclear envelope. Together, these are referred to as the perinuclear theca (PT), which covers the external surface of the nuclear envelope [4]. The PT contains specific cytoskeletal proteins [5,6] and is sandwiched between the inner acrosomal membrane and nuclear envelope, while it caudally resides between the plasmalemma and the nuclear envelope [7,8]. Moreover, PT is mainly comprised of actin [9,10], calicin [11] or superoxide dismutase (CuZn-SOD) [12]. Each of these PT proteins have a specific function that leads to the correct sperm functionality during spermiogenesis, as well as the establishment and maintenance of sperm head architecture [7,8], the capacitation and acrosome reaction process [9,13] or even a defense against cell damage mediated through superoxide anion radicals [12,14,15]. However, abnormalities of these transformations during spermiogenesis could lead to functional deficiencies, such as alterations of chromatin condensation, susceptibility to DNA damage, presence of an abnormally small acrosome or immaturity of the sperm plasma membrane [16,17,18,19,20]. Furthermore, it could lead to abnormal morphologies, such as the appearance of nuclear vacuoles [16,20,21,22,23].

More than sixty years ago, the nuclear vacuoles were studied using transmission electron microscopy (TEM) [24,25,26] and described as irregular cavities in the nucleus of late spermatids, forming the so-called head vacuoles that were a conspicuous feature of human spermatozoa [24]. Nuclear vacuoles are relatively common (>95% of spermatozoa contain vacuoles) in both fertile and infertile men [20]. Additionally, nuclear vacuoles in head spermatozoa largely vary in number (one or more) [22,27,28,29], shape-size (small or large) [16,21,30] and position (anterior, middle or posterior) [20,22,28,29]. However, the origin and the functional role of nuclear vacuoles in male infertility remains largely debated [20,31,32]. Early reports have proposed that sperm-head vacuole derive from acrosomal content [33,34], pocket-like nuclear concavities [16,18,21,30] or nuclear indentations packed with membranous material [31]. Moreover, sperm vacuoles are also linked to male fertility potential. For instance, the presence of human sperm vacuoles is associated with implantation failures or miscarriages [35,36,37,38,39], failure of sperm chromatin condensation at replacing histones by protamines [18,19,21,31,40] or DNA fragmentation [18,41]. Conversely, different studies have also argued that nuclear vacuoles result from a natural physiological process unrelated to DNA fragmentation [34,42], not affecting the sperm quality parameters [28,42] and/or sperm head morphology [42]. Moreover, spermatozoa with vacuoles were observed even after selection by swim-up and density gradient centrifugation [32,43].

Sperm morphology has been a debated topic for many years, and normality values were modified in different World Health Organization (WHO) manuals from 1980 to 2010 [44,45,46,47,48], lowering the threshold from 80.5% to 4%. These normalized changes have also affected the estimation of the vacuole sizes, since they could be more accurately analyzed due to the incorporation of MSOME (motile sperm organelle morphology examination), using Nomarski differential interferential contrast (DIC) microscopy [49], and thereby providing new standards to define abnormal sperm morphologies. Likewise, there is still no clear consensus on whether morphology affects reproductive success [48,50]. MSOME techniques [49] improve high-resolution details (up to ×6000) concerning the biological composition of sperm-head vacuoles [20] and provided descriptions on the presence of small and large sperm-head vacuoles [16,21], establishing a model of large vacuoles in a variable range between 4–50% [16,17,23,27,28,29,51]. However, the classifications used to define their sizes seem to be contradictory due to the different published results [20]. Nonetheless, MSOME mainly focuses on external sperm shape features [31], and ultrastructural morphometrics of sperm-head vacuoles remain poorly characterized using other techniques such as TEM [19,31,52], confocal microscopy [19] or atomic force microscopy [16,53]. Furthermore, the exact content of these vacuoles has not been previously identified [31]. Here, we develop a combined novel approach using TEM and immunogold labelling to describe the incidence, position, size-related variation and molecular content of human sperm-head vacuoles in order to elucidate their origin and physical–chemical nature.

## 2. Results

The presence of human sperm vacuoles was observed in all sperm samples from 17 donors. A total of 1908 sperm cells from all donors were analyzed. The incidence of sperm-head vacuoles was 16.8% to 70.7% (47.7 ± 15.7%; mean ± standard deviation) from total analyzed sperm, with a mean incidence of 1.3 ± 0.2 vacuoles per sperm head (Table 1). The vacuole frequencies significantly decreased from tip to posterior sperm head regions (Figure 1a). Vacuoles were mainly located at the tip of the sperm heads (81%; 68.2–96.9, in range) compared to the middle (13%; 0–28.6, in range) and posterior (6%; 0–18.2, in range) regions (Mann–Whitney test *p* < 0.05, respectively).

Moreover, the relative vacuole area (RVA) was present in 1.1% to 11.5% (9.7 ± 2), and a positive significant correlation (*r* = 0.49; R^2^ = 0.24; *p* < 0.001; *n* = 270 sperm sections) was detected between sperm vacuole and nucleus areas (Figure 1b). Therefore, it could be deduced that as vacuoles increased in size, greater nuclei were observed and were able to explain ≈25% variability of sperm nucleus sizes.

Otherwise, ultrastructural morphological analyses allowed us to observe when the nuclear envelope of human spermatozoa suffered an invagination process (Figure 2a–f). In addition, invaginations were observed inside the nucleus surrounded by the nuclear envelope (Figure 2g,h). These invaginations were mainly located at the tip region of the nucleus (Figure 2a,b). However, the shape and structure of the acrosomes were not altered by the invaginations (Figure 2b,d). Thus, the content of nuclear invaginations seemed to derive from SAR-PT, the thin layer of the cytoplasm present in the sperm head (Figure 2f). Moreover, when immunocytochemistry and TEM analyses were combined, we found that actin (Figure 3a,b), calicin (Figure 3c,d) and CuZn-SOD (Figure 3e,f) were present inside the nuclear vacuoles, and in the perinuclear theca (Figure 3c,e). Furthermore, control experiments exhibited high specificity of antibody labeling. Background control samples were devoid of gold particles and no counts were performed on the corresponding human sperm (see Appendix A). According to our results, we propose that the vacuoles originate from nuclear-envelope invaginations and enclose specific cytoskeletal elements from the SAR-PT (Figure 2f).

## 3. Discussion

A combined technical approach using TEM and immunocytochemistry has allowed to accurately examine, for the first time, the position, morphology, and molecular characteristics of nuclear vacuoles from human sperm heads. Our findings demonstrated that human sperm vacuoles are related to nuclear-envelope invaginations. The ultrastructural analyses revealed the presence of different cytoskeletal proteins in the vacuoles, and an important intracellular enzyme (CuZn-SOD1) involved in the protection of spermatozoa from superoxide anion radicals [12,14]. The presence of these proteins both inside the vacuoles and the cytoplasm contained in the perinuclear theca, where these proteins are usually contained, supports their possible functional importance within the sperm head, and even moreso the vacuole origin from intranuclear cytoplasmic invaginations. Indeed, the presence of actin and calicin in the head of the selected spermatozoa has been confirmed previously in several studies, although the precise distribution is controverted. The methods used in the previous reports are different (fixation, antibody specificity, immunofluorescence, immunogold, Western blot, etc.) [9,10,11]. Therefore, here we combined TEM microscopy with immunogold techniques to identify with precision the presence of these cytoskeletal proteins. The polyclonal anti-actin and anti-calicin antibodies used for this study allowed us to immunolocalize the low presence of these antigens in the ultrathin sperm sections. These specific and low-density gold particles found in this sperm section could be due to the small volume of cytoplasmic material contained in nuclear invaginations. Moreover, we used negative control, by omitting the first antibody, to ensure their specificity. We show that gold particles were absent in these controls, so there was no non-specific labelling.

Regarding the presence of nuclear vacuoles, our TEM results revealed in spermatozoa from normozoospermic subjects a variable presence of vacuoles in accordance with previous reports [22,28,51]. However, the incidence of vacuoles in these cells was lower compared with previous estimations [22,27,28,51]. Discrepancies in vacuole densities are probably due to technical differences, since light microscopy (LM) was used in early reports, i.e., [16,21,31], which defined the vacuoles as surface concavities or lighter translucent areas of variable sizes. In addition, LM produces pseudo-3D images derived by the refraction of light passing through the sample of different thickness and optical density. Therefore, LM assays could detect surface irregularities of the sperm and be confused with vacuole presence [31]. Furthermore, LM is not able to provide the detailed information required to characterize ultrastructural morphological features derived from TEM techniques [52,54]. However, and as a limitation of our study, it should be noted that TEM is a sophisticated, time-consuming and expensive technique. Even with the addition of an immunocytochemical study, it is very difficult to analyze a high number of samples with different pathologies using TEM. Here, we used only normozoospermic samples (as normal quality according to WHO, 2010), to non-induce sperm variability with other seminal quality.

In accordance with previous studies, our data reveal the presence of 1–2 vacuoles per spermatozoon, which have also been detected in both normal and altered semen parameters [20,22,27,28]. Furthermore, regarding the vacuole positions in the sperm head and according to previous findings [22,28,29], a significantly higher incidence was detected at the tip of the sperm head compared to the middle and posterior regions. However, contrary to our findings, other authors found a homogeneous distribution at the tip [28] or middle region [22], or even commented that almost all the vacuoles were detected at the tip and middle region [29]. Discrepancies are derived from the methods used since the TEM allowed us to isolate and analyze from longitudinal sections of sperm heads. Because of this, it could be possible that many sperm head vacuoles selected as tip regions could correspond to middle regions. Therefore, based on our results, the vacuole range determined was derived from the two-thirds part of the anterior region of the sperm heads.

Regarding the size of the sperm head vacuoles, most studies have qualitatively dichotomized the vacuoles as ‘small’ and ‘large’ in size [16,18,19,20,28,29,36,51] or without a specific size criterion [17]. Other studies have determined the RVA size [19,22,27], showing that RVA values higher than 13% are mainly present in abnormal sperm associated with chromatin disorders. Our results from normozoospermic subjects agree with RVA counts below that threshold. Likewise, no differences between morphologically normal spermatozoa and spermatozoa with large vacuoles were detected [55], and the presence of vacuoles did not allow differentiation between spermatozoa from fertile and non-fertile men [22]. The findings suggest that, although there is no consensus about the optimal technique for the study of sperm-head vacuoles or even a standardized criteria for classifying their size [16,17,18,19,20,28,29,36,51], RVA is an appropriate and reproducible parameter for their diagnosis [19,22,27]. Therefore, according to previously mentioned models [19,22,27] and our results in normozoospermic samples, a vacuole could be considered a normal sperm structure when its relative area does not exceed 13% of the nucleus area (vacuole + chromatin areas).

Moreover, despite the correlation published in the literature between large vacuole area and chromatin decondensation, DNA fragmentation and abnormal sperm head [19,20,21], no relationship was found by different authors between abnormal head occurrence and DNA fragmentation [22,42]. These studies showed that vacuoles are equally present in spermatozoa with normal and abnormal heads [20]. These data are corroborated by the fact that the presence of vacuoles is not related to the main factors affecting fertility, such as sperm concentration and motility, as well as live birth rate. Based on this evidence and our results in normozoospermic subjects, sperm vacuoles may not be associated with pathological features in sperm quality.

Finally, our findings demonstrate that vacuoles are invaginations of the nuclear envelope and contain different proteins that may be involved in sperm physiological processes linked with their formation, protection, motility, and capacitation, and therefore contrary to the studies that consider an acrosomal origin [33,34]. Moreover, an increase of the sperm nucleus area may be due to an increase of the area of the vacuole; thereby, the sperm morphology observed does not affect sperm quality [42]. This may indicate that vacuoles, only present in human sperm [56,57], even from normal and abnormal heads [42] and in both fertile and non-fertile men [20,22], could be considered physiological structures of human sperm [22,28,31,34]. Thus, based on our findings, we propose to replace the term ‘nuclear vacuoles’ with ‘nuclear invaginations’, since the nuclear envelope causes this structure.

## 4. Materials and Methods

### 4.1. Semen Samples Preparation

Semen samples were collected by masturbation from 17 normozoospermic donors (Hospital “La Fe”; Valencia, Spain) after three to four days of sexual abstinence. Sperm concentration, motility and morphology parameters were determined in accordance with WHO (WHO, 2010). After semen liquefaction, semen samples were washed 2 times with phosphate-buffered saline (PBS) buffer to remove seminal plasma at room temperature (RT). Each semen sample was fixed and processed following 2 different protocols prepared for morphological and immunocytochemistry analysis. This research was approved by the Bioethics Committee of the Universidad de Alicante (UA-2021-07-13) according to the principles of the Declaration of Helsinki principles. An informed consent form was accepted and signed from each donor.

### 4.2. Electron Microscopy

For morphological examination, semen samples were centrifuged (600× *g*, 5 min), supernatants carefully removed and pellets fixed in a mix of 2% *v*/*v* glutaraldehyde (Electron Microscopy Sciences, Hatfield, PA, USA) and 1% *v*/*v* paraformaldehyde (Electron Microscopy Sciences, Hatfield, PA, USA) in PBS buffer following the described protocols [31] and incubated at 4 °C for 1 h. Samples were then embedded in 2% *w*/*v* agarose (A6877 Sigma-Aldrich, St. Louis, MO, USA) in PBS buffer, postfixed with 1% *v*/*v* osmium tetroxide (Electron Microscopy Sciences, Hatfield, PA, USA) and dehydrated [41] and embedded in EPON resin (EMbed-812, Electron Microscopy Sciences, Hatfield, PA, USA). Ultrathin sections were counterstained with uranyl acetate 5% and lead citrate 2.5%. For immunocytochemistry examination, semen samples were centrifuged (600× *g*, 5 min), supernatants carefully removed and pellets fixed in 1% *v*/*v* paraformaldehyde in PBS and incubated at 4 °C for 1 h. Samples were then embedded in 2% *w*/*v* agarose in PBS buffer, and dehydrated. After that, semen samples were embedded in LR White resin (London Resin Company, London, UK).

### 4.3. Immunocytochemical Study

Colloidal gold particles were used as a marker for cytochemistry at the ultrastructural level, and a two-step method was used [58]. The grids were floated on a drop of the following primary antibodies, separately: anti-actin polyclonal antibody from rabbit (Sigma-Aldrich, St. Louis, MO, USA), diluted 1:20; anti-calicin polyclonal antibody from goat (Santa Cruz Biotechnology Inc., Santa Cruz, CA, USA), diluted 1:10; and anti CuZn-SOD1 (copper- and zinc-containing superoxide dismutase) monoclonal antibody from mouse (Sigma-Aldrich, St. Louis, MO, USA), diluted 1:100. The three antibodies were diluted in PBS supplemented with 0.5% BSA for 1h. After washing, the grids were floated on a drop of protein A conjugated with colloidal gold of 10 nm mean diameter (British Biocell International, Cardiff, UK) diluted 1:60 in PBS buffer for 1 h. After washing in PBS buffer and in distilled water, the grids were counterstained with uranyl acetate and lead citrate. The specificity of the used antibodies was tested by omitting the anti-actin, anti-calicin and anti-CuZn-SOD1 antibodies during immunocytochemical analysis. One of the criteria for choosing polyclonal antibodies is because these proteins have been shown in the literature to be present with monoclonal antibodies which are comparable to those described in the previous studies [11,59,60,61]. Another criterion is that the presence of antigens is lower using ultrathin sections in transmission electron assays than entire cells using immunofluorescence.

### 4.4. Sperm Head Vacuole Count and Morphological Analysis

Ultrathin sections were observed using TEM (JEM-1400 Plus, JEOL Ltd., Tokyo, Japan) with a voltage of 120 kV and micrographs acquired at 2000× magnification with a Orius SC200 (GATAN Inc., Pleasanton, CA, USA) camera. Sperm vacuoles were analyzed from ≈100 spermatozoa longitudinal sections from each donor without sorting their sizes [16,21]. Total sperm sections were analyzed together. The incidence of vacuolated spermatozoa was recorded. In addition, based on these TEM micrographs, the vacuoles’ position on the head (tip, middle and posterior region) was evaluated as previously described [29]. Additionally, the vacuole number per spermatozoon and the relative vacuole area (RVA, in %) were determined [27] as follows: [vacuoles area (µm^2^)/nucleus area (µm^2^)] × 100. Other morphological parameters, such as the vacuole and nucleus area (vacuole + chromatin) in square micrometers (µm^2^) of 15 to 20 sperm longitudinal sections from each donor (*n* = 270) were measured using the SigmaScan^®^ Pro 5 (SPSS^TM^, Chicago, IL, USA) digital imaging software.

### 4.5. Statistical Analyses

The Shapiro–Wilk test showed significant differences in distribution and variance for vacuole counts (W = 0.283–0.626, *p* < 0.001). The two-tailed Mann–Whitney U test was then used to test percentage differences within vacuole positions from sperm heads. In addition, an Ordinary Least-Squares (OLS) regression was performed to estimate the relationship between log-transformed sperm vacuole areas on nucleus areas. Descriptive (mean ± standard deviation; SD) and statistical procedures were conducted using SPSS^®^ Statistics 22.0 (IBM^®^, Armonk, NY, USA). The threshold for statistical significance was set to *p* < 0.05.

## 5. Conclusions

The vacuoles in sperm heads are invaginations derived from the nuclear envelope. Moreover, they contain sperm-specific cytoskeletal proteins from perinuclear theca. Their incidence is abundant in sperm from normozoospermic subjects and mainly located at the tip of the head of the spermatozoon. According to our findings, sperm-head vacuoles are nuclear concavities with a common structure, which could have a physiological function instead of being a morphological alteration of human sperm. However, the function of sperm with invaginations was not analyzed in this study; further studies are needed to know their physiological function, correlation with sperm parameters, seminal quality and success in assisted reproduction.

## Figures and Tables

**Figure 1 ijms-24-10027-f001:**
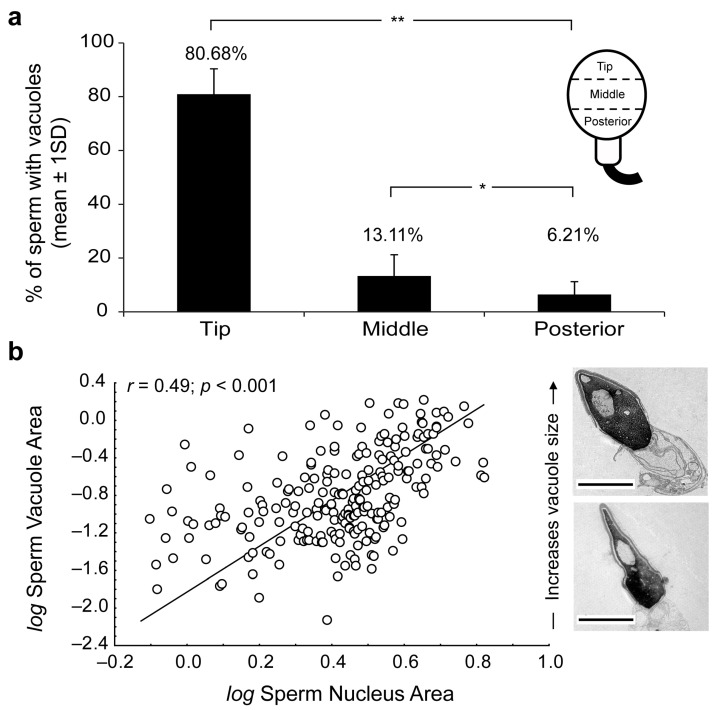
(**a**) Percentage of vacuoles present at the tip, middle or posterior regions of sperm heads. Significant Mann–Whitney U-test differences at *p* < 0.05 (*) and *p* < 0.01 (**). (**b**) Scatterplot of log-transformed sperm vacuole area against nucleus area (vacuole + chromatin). Note the strong positive association between metrics (in μm^2^). Significance of Pearson correlation coefficient (Pearson’s *r*; *p* < 0.05) derived from Ordinary Least-Squares (OLS) regression is shown. Transmission electron microscopy (TEM) micrographs depict size-related variation in sperm-head vacuoles. Scale bar 2 micrometers (μm).

**Figure 2 ijms-24-10027-f002:**
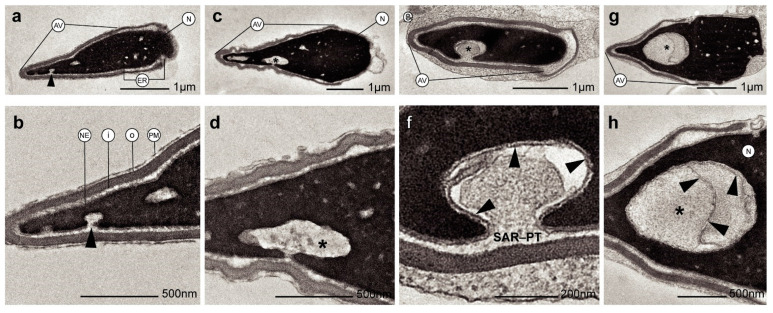
Transmission electron microscopy (TEM) micrographs of longitudinal sections of human spermatozoa showing different invagination process from nuclear envelope inside the nucleus. (**a**,**b**) Human sperm head with a small invagination of nuclear envelope (see arrowhead). (**c**,**d**) Larger invagination of nuclear envelope (asterisk). (**e**) The invagination of nuclear envelope (asterisk) at greater magnification (**f**) denotes that the origin of nuclear envelope’s invagination (arrowheads) comes from SAR-PT. (**g**,**h**) Sections showing an invagination of nuclear envelope (asterisk) integrated within the nucleus (arrowheads show nuclear envelope). Acrosomal vesicle (Av); equatorial region (ER); nuclear envelope (NE); plasma membrane (PM); inner acrosomal membrane (i); outer acrosomal membrane (o); subacrosomal region of perinuclear theca (SAR-PT); nucleus (N).

**Figure 3 ijms-24-10027-f003:**
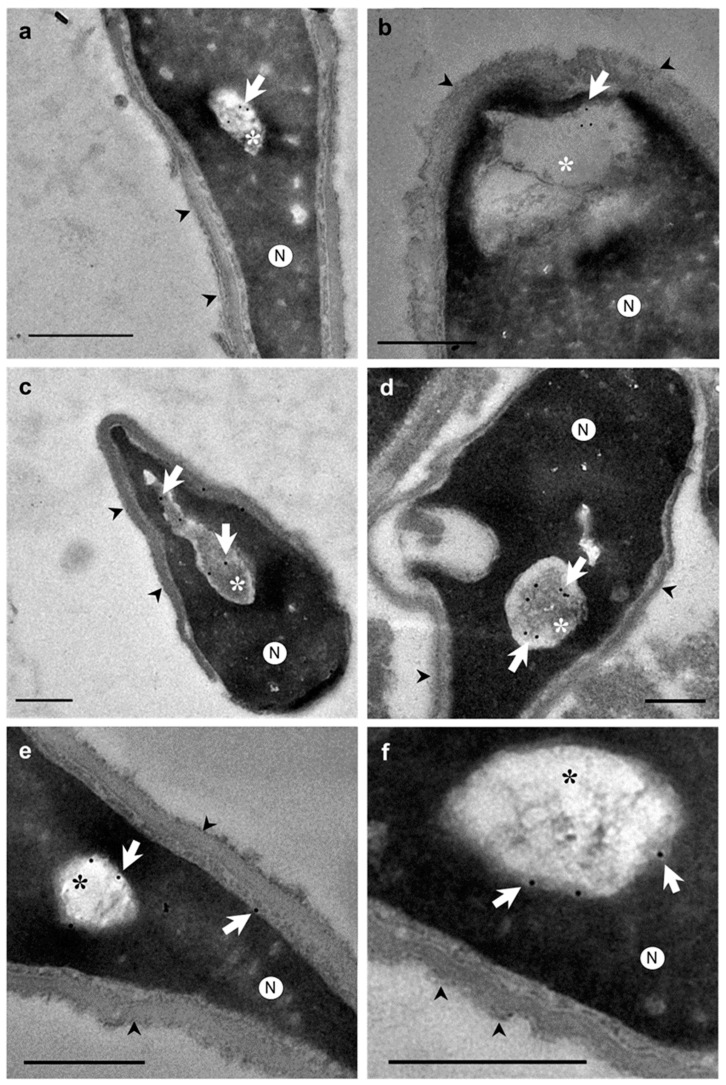
Transmission electron microscopy (TEM) micrographs of longitudinal sections of human spermatozoa showing immunogold labeling. Arrowheads indicate intact acrosome. Asterisks indicate the nuclear sperm vacuoles. (**a**,**b**) Polyclonal anti-actin antibody; actin protein is present in nuclear vacuoles (arrows). (**c**,**d**) Polyclonal anti-calicin antibody; calicin protein is present in nuclear vacuoles (arrows). (**e**,**f**) Monoclonal antiCuZn-SOD1 (superoxide dismutase) was used to perform this assay, CuZn-SOD enzyme is present in nuclear vacuoles (arrows) and subacrosomal region of perinuclear theca. These results show that vacuole content is cytoplasm in origin with cytoskeletal and enzyme proteins. Nucleus (N). Scale bar 500 nm common to all micrographs.

**Table 1 ijms-24-10027-t001:** Incidence and morphometrics of sperm-head vacuoles from normozoospermic samples.

Sample	Sperm Cells (*n*)	Vacuolated Sperm (%)	Vacuoles/Cell (*n*)	Relative Vacuole Area (RVA) ^a^
1	101	42.574	1.615	9.659
2	96	64.583	1.455	9.204
3	124	43.548	1.238	10.081
4	98	51.020	1.391	7.365
5	170	24.118	1.167	1.144
6	106	58.491	1.455	9.310
7	83	48.193	1.391	7.826
8	75	30.667	1.294	10.185
9	167	16.766	1.409	11.379
10	134	38.060	1.150	8.893
11	100	65.000	1.235	9.160
12	106	48.113	1.500	6.540
13	109	34.862	1.100	11.039
14	106	45.283	1.118	10.119
15	106	70.755	1.200	11.370
16	104	62.500	1.667	11.550
17	123	66.667	1.391	6.359
	Mean	47.718	1.340	9.658
	SD	15.678	0.170	1.978

^a^ RVA: relative vacuole area (%) = [vacuoles area (μm^2^)/nucleus area (μm^2^)] × 100. Nucleus area refers to vacuole + chromatin (see Materials and Methods section; for further details). Standard deviation (SD).

## Data Availability

For further information about the data presented in this study, contact the corresponding author.

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
