# Peer review of "Human Sperm Head Vacuoles Are Related to Nuclear-Envelope Invaginations"

_ijms, 2023, doi:10.3390/ijms241210027_

Round 1

Author Response

Please see the attachment,. Our answer is written in red

Reviewer 2 Report

By using TEM authors show that the so-called vacuoles present in the spermatozoon nucleus of normozoospermic subjects simply are portions of cytoplasmic invaginations of the nuclear envelope. This might be considered obvious and of low interest; on the contrary of a certain interest is that, at difference from the past, several molecules contained in these vacuoles have been identified by a combined novel approach by using TEM and immunogold labelling. 

I suggest to remark in the results-discussion-conclusion that the gold particles are present either in the “nuclear vacuoles” and in the “perinuclear theca”, thus confirming that the content of the vacuoles is the same cytoplasm of that of the perinuclear theca.

Fekonia et al. Biomed. Res. Int. 2014, 927841, published TEM images good as yours showing that in normospermic men the spermatozoon head vacuoles are large nuclear indentations of various sizes and positions, packed with membranous material organized in membrane whorls. Have you seen these same images? Have you considered them in your quantitative analysis?

 Pay attention to English language and some unclear sentences, some of which are: 

1.     Pag.1, row 46. In the sentence “The PT contains specific cytoskeletal proteins [5,6] and sandwiched between the inner acrosomal….”,  something is lacking, perhaps “and is”

2.     Pag.7, row 173 . “We show that of gold particles were absent …. “.  Delete “of”

3.     Pag.7, rows 183-184.  The sentence  “LM assays could detect surface irregularities of the sperm and confused with vacuole presence [31].” needs correction

4.     Pag.7, rows 187-189, unclear, need correction

           5.     Pag.10, row 312.  “phycological function “  what is “phycological”?

Pay attention to English language and some unclear sentences

Author Response

Please see the attachment. Our answer is written in red.

Reviewer 3 Report

The manuscript of Gómez-Torres et al. entitled “Human Sperm Head Vacuoles are Related to Nuclear-Envelope Invaginations” is a morphological analysis of vacuoles localized in the human sperm head. The AA use ultrathin sections to report the localization of emerging vacuoles in the sperm head and post-embedding techniques to localize some antigens of interest. Since the presence, nature, function, and significance of vacuoles in the sperm heads is controversial, this paper has a potential interest. However, I have some observations.

Firstly, and more important, the observations the AA reported are not news and there are several papers in the literature dealing with these aspects. The AA justify their work with a better resolution at the EM level of the vacuole structure and with a better localization of the antigens by immunocitochemistry. In my opinion the AA should make clearer their novelties and the differences between their observations and already published data.

Fig. 3a,c. My curiosity: I seem to see scattered gold particles outside the nucleus, but they look much clearer than those inside the nucleus. It's correct?

Fig. 3c I can see four particles inside the nuclear vacuole and four particles outside. How can I say that the localization of the antigen is only within the vacuole? Better and clear in d.

Fig. 3e. “CuZn-SOD enzyme is presented in nuclear vacuoles (arrows) and subacrosomal region of perinuclear theca” I know that pre-embedding technique does not often allow you to have many particles, but I can see only one gold particle on the perinuclear theca. Therefore, I don't think it's sufficient to define this specific location unless the AA have other images.

Fig. 3. Please add arrowheads in the legend.

Why the AA chose protein A as preferred method and not usual secondary gold-conjugated antibodies? Protein A has strong affinity for rabbit, but weak affinity for goat and mouse.

“According to our findings, sperm head vacuoles are nuclear concavities with a common structure, which could have a physiological function instead of being a morphological alteration of human sperm”.

What possible function the AA retain??

Author Response

(The authors gave the same response as above.)

Reviewer 4 Report

Entitle manuscript “Human Sperm Head Vacuoles are related to nuclear-envelope invaginations” by María José Gómez-Torres et al., provide in depth data by analyzing TEM results of normozoospermic human samples. Data presented in this manuscript are highly significant and revealed a new feature of nuclear functionality of sperm head that could play its role during acrosome reaction. There is a need for language revision for clarity of sentences and a minor revision for further action. Comments are mentioned below for suggestion,

·         Generally, there are various types of vacuoles based on cellular activities. But here what was the selection criteria of vacuole that was measured?  Did the author define the ranging or morphological features of vacuole.

·         The term “Top” in the sperm head as mentioned in the manuscript should be replaced with anterior region.

·         Did the author notice the vacuole invagination at the middle or posterior region of sperm head?

·         The author needs to clearly indicate throughout the manuscript that “Vacuole originates from SAR-PT and nuclear envelope invaginates”. However, in some parts of manuscript (abstract, result and conclusion) it was confusing.

·         What “N” designate in the Figure-2A, C & H.

·         No supplementary results were found.

Pleas review the main comments

Author Response

(The authors gave the same response as above.)

Round 2

Reviewer 2 Report

Minor corrections are needed

1.     Legend to Figure 3. Rows 150,151,153: present, not presented. correct . 

2.     Discussion rows 163-165. Correct the sentence “Due---sperm head. The presence..” as follows: The presence of these proteins both inside the vacuoles and the cytoplasm contained in the perinuclear theca, where these proteins are usually contained, supports a their possible functional importance within the sperm head, and even more the vacuole origin from intranuclear cytoplasmic invaginations. Indeed, the presence…..

Author Response

We respond you in red letters.

  1. Legend to Figure 3. Rows 150,151,153: present, not presented. correct . 

Modified. Thank you very much for your assistance.

  1. Discussion rows 163-165. Correct the sentence “Due---sperm head. The presence..” as follows: The presence of these proteins both inside the vacuoles and the cytoplasm contained in the perinuclear theca, where these proteins are usually contained, supports a their possible functional importance within the sperm head, and even more the vacuole origin from intranuclear cytoplasmic invaginations. Indeed, the presence…..

Thank you very much. This structure is clearer than my sentence.

Reviewer 3 Report

The authors sufficiently answer my questions, although, I disagree with the answer  to my observation of Fig.3c. The AA claim in the explanation of Fig. 3c “Polyclonal anti-calicin antibody, calicin protein is presented in nuclear vacuoles (arrows)” but I can see other gold particles outside of the nucleus (I put dark arrows). However, the Authors are responsible for their statements and I think the manuscript can be now accepted.

Author Response

Thank you very much for your comments and we express our gratitude for your assistance and observations. But we cannot see that dark arrows that you were put. Moreover, in the Fig. 3c, we only can see four smaller spots (two on the left and two on the right) outside the sperm cell. Obviously, the size of those spots is smaller than 10 nm, so those are not gold particles.